# Population dynamics of sugarcane borers, *Diatraea* spp., under different climatic scenarios in Colombia

Julián Andrés Valencia Arbeláez[1]*, Alberto Soto Giraldo[2‡], Gabriel Jaime Castaño Villa[3‡], Luis Fernando Vallejo Espinosa[2‡], Melba Ruth Salazar Gutierrez[4‡], Germán Vargas[5]

**1** Facultad de Ciencias Agrícolas, Corporación Universitaria Santa Rosa de Cabal, Santa Rosa de Cabal, Risaralda, Colombia, **2** Facultad de Ciencias Agrarias, Universidad de Caldas, Manizales, Caldas, Colombia, **3** Departamento de Ciencias Naturales, Universidad de Caldas, Manizales, Caldas, Colombia, **4** Biological Systems Engineering Department, Washington State University, Prosser, Washington, United States of America, **5** Centro de Investigación de la Caña de Azúcar de Colombia, Florida, Colombia

‡ These authors also contributed equally to this work.
* julian.valencia@unisarc.edu.co

**Data Availability Statement:** All files are available from the OSF database (https://osf.io/ms39c/).

## Abstract

Seasonal temperature and precipitation patterns on a global scale are main factors to which insects and plants adapt through natural selection, although periodic outbreaks in insect populations may occur in areas where they had not been previously reported, a phenomenon considered as a consequence of global warming. In this study, we estimate the distribution of sugarcane borers, *Diatraea* spp., under different climate scenarios (rcp26, rcp45, rcp60 and rcp85.) Insects were collected weekly in four sugarcane fields from four different towns in the department of Caldas (Colombia) during 2017, and also in several sugarcane fields in the Cauca River Valley (CRV) between 2010 and 2017. The influence of climatic variables on different agro-ecological zones of the CRV sugarcane fields was defined by climatic data between 2010 and 2017 (maximum and minimum daily temperatures, and accumulated precipitation). The estimate of an optimal niche for *Diatraea* spp. includes temperatures between 20˚C and 23˚C, accumulated annual rainfall between 1200 and 1500 mm, dry months with precipitations below 50 mm, slopes of less than 0.05 degrees, crop heterogeneity with an index of 0.2 and primary production values of 1.0. Data suggests *Diatraea* population is considerably influenced by adverse climate change effects, under the premise of an increase in local and global temperatures, reducing its population niches as well as the number of individuals.

## Introduction

The Earth's temperature has increased 1.8˚C over the past 35 years and the last two decades have been among the warmest since temperatures began to be recorded. Since the early 80's, tons of different crops have been lost annually due to global warming [1, 2]. In some cases,

**Funding:** The Colombian Institute for the Development of Science and Technology (COLCIENCIAS) provided funding for this study through grant program number 647. The Colombian Sugarcane Research Center (Centro de Investigación de la Caña de Azúcar de Colombia, Cenicaña) provided support for this study in the form of use of infrastructure and laboratories.

**Competing interests:** The authors have declared that no competing interests exist.

however, losses were compensated by higher yields achieved through genetically improved crops and other technological advances [3]. High seasonal temperatures may become widespread in some areas of Central and South America during the remainder of the century [4], and temperatures are estimated to increase by 0.4°C to 1.8°C by 2020, being more pronounced in tropical areas. High temperatures (especially when increases exceed 3°C) may significantly affect agricultural productivity, representing serious impact to commodity crops for food-insecure populations, with very negative scenarios for some crops more than for others [1, 5].

Increases in pest incidence have been observed in agricultural systems, associated with climate change events, such as prolonged droughts, hurricanes, heavy and out-of-season rains, among others. Changes in pest dynamics often pass unnoticed, overshadowed by human disasters caused by such climatic events. And yet, pest outbreaks contribute to incremental losses, forcing the use of pesticides that usually fail to solve the problem [6, 7].

Seasonal patterns of temperature and precipitation are known to be the main factors determining organisms distribution at a global scale [8, 9]. Insects and plants adapt to combinations of these factors through natural selection, although periodic outbreaks of insects have been documented to occur especially in physically stressed areas as a consequence of global warming [7]. Due to the response of insect populations to changes in local climate conditions, fluctuations in these populations may serve as indicators of local or global climate change [9]. Temperature may thus be regarded as the most important variable in insect response, considering its role on biochemical processes and the insect's dependence on environmental temperature to thermoregulate, thereby affecting infestation dynamics [6, 9–11]. Infestation dynamics are more notorious with pronounced climate variability events such as El Niño and La Niña [4, 12, 13]. Consequently, the combination of these factors enables the development of agrometeorological indices to identify areas of greater and lower risk of climate variability, and the provision of regional recommendations for future climate variation events [14].

Monitoring studies with a bio-indicator of climate change can play a role in at least two main fronts: data collected can be used to predict climate change effects and long-term monitoring can detect changes in abundance and distribution, an important input in redirecting management strategies to establish appropriate control measures in advance. Additionally, data collection and monitoring constitute valuable sources of information on the effects of climate change in tropical latitudes, where information is insufficient.

The effects on biomass production and sucrose content [15] cause by species belonging to the *Diatraea* genus (Lepidoptera: Crambidae) are considered of great economic importance to sugarcane in the Americas. The damage caused by these sugarcane borers include destruction of buds in planting material; injury to seedlings, resulting in the so-called 'dead heart'; and perforations to fully developed stalks, reducing internode growth and allowing the propagation of microorganisms that decrease sucrose content [16]. Even though a great number of studies on biology and integrated pest management are available, information is lacking on insect's ecology and population dynamics, particularly in relation to climate change effects. To understand how climate change can affect sugarcane borers distribution, predictive inter- and intra-annual scale models were applied at the global, national and local scales, aiming to develop a framework to guide pest management decisions, thus reducing levels of uncertainty in the face of different climate scenarios.

## Materials and methods

Insects were collected in western Colombia in the departments of Caldas, Risaralda and Valle del Cauca. In Caldas, insects were collected during 2017 from fields in four municipalities (Riosucio, Supía, Filadelfia, Neira and Manzanares); whereas in the departments of Risaralda

and Valle del Cauca, samples were collected from 2010 to 2017 from different municipalities in what it is known as the Cauca River Valley (CRV), a region that extends from northern Cauca to southern Risaralda (Fig 1). Larvae were collected in each sugarcane plot by a uniform 2 man-hour sampling effort, following the linear distribution of crop rows. Larvae were then taken to the laboratory and identified following taxonomic keys [17]. In addition, this information was used in combination with the CABI—Invasive Species Compendium (www.cabi.org)–to increase data on distribution of *Diatraea* spp. and increasing modeling accuracy.

Climate data (daily precipitation; maximum, minimum and daily temperatures) between 2010 and 2017 were obtained from the 37 weather stations of the Colombian Sugarcane Research Center (*Centro de Investigación de la Caña de Azúcar de Colombia*, *Cenicaña*) located across the CRV (Fig 1). The distribution map of the weather stations was generated by QGIS 3.6.1, according to coordinates of the different stations.

Missing weather data was generated using Marksim⑧, which provided daily scale weather data files (http://gismap.ciat.cgiar.org/MarkSimGCM/). At the same time, open-access weather portals were consulted such as WorldClim, from which 19 bioclimatic variables were collected on monthly rainfall and temperature [18]. The variance inflation factor (VIF) was calculated to analyze correlations between variables [19, 20], providing an index that measures the extent to which the variance (the square of the estimated standard deviation) increases due to the collinearity, which allows for discarding climatic variables that could generate noise through redundancy (Eq (1)).

$$VIF_i = \frac{1}{1 - R_i^2} \qquad (1)$$

Where $R_i^2$ is the determination coefficient of the regression equation

Following the VIF analysis, only 6 of the 19 variables provided by WorldClim were used, plus slope (topographic slope in degrees), slope-aspect (which is defined as the compass direction to which a slope faces measured in degrees), altitude (meters), vegetation index (VI), terrestrial primary production (TPP, gCm$^2$ d$^1$), and TH24 (topographic heterogeneity calculated for the 24 surrounding cells) [21, 22], and used along with those supplied within the ModestR⑧ software (http://www.ipezes/ModestR) [23] (Table 1).

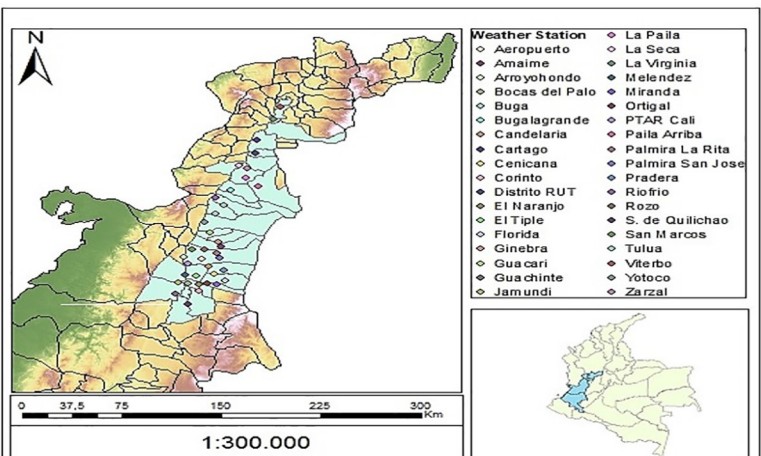

**Fig 1. Cenicaña's weather stations in the Cauca River Valley (CRV), Colombia.**

The global climate model (GCM) BCC_CSM1 from the Beijing Climate Centre was used to test the climatic variables. The GCM is a fully coupled global climate and carbon model that includes the interactive vegetation and global carbon cycle, the oceanic component, the terrestrial component, and the sea-ice component. In addition, according to the Colombian Institute of Hydrology, Meteorology and Environmental Studies (*Instituto de Hidrología*, *Meteorología y Estudios Ambientales*, IDEAM) this model fits well Colombian meteorological conditions [24]. The exchange of atmospheric carbon with the terrestrial biosphere was calculated at each stage of the model (20 min) and representative concentration pathways (RCP) were used. These four paths of greenhouse gas concentration (not emissions), adopted by the International Panel on Climate Change (IPCC), are used to model climate by describing four possible future climate scenarios, based on the amount of greenhouse gas emissions projected for the coming years. These representative concentration pathways were referenced as rcp26, rcp45, rcp60 and rcp85 [25].

ModestR® was used to generate distribution maps and analyze the effect of climate variables on *Diatraea* spp. distribution. ModestR® is an R-based program that integrates a combination of statistics, maximum entropy and Bayesian methods to estimate probable distributions, subject to restrictions imposed by the environment [26, 27]. The model requires only presence-absence data, and uses continuous and categorical variables, incorporating interactions between different variables, allowing assessing the role of each environmental variable, while over-adjustment is avoided using regularizations. The output variable enables making distinctions among areas (only discrete distinctions however) and can be used in multiple applications and at all scales [28].

RWizard® was used to calculate the variable effects of richness, niche, and construction. The packages used were FactorsR, to identify factors affecting species richness; EnvNicheR, to estimate the niche of multiple species and environmental conditions favoring greater species richness; and SPEDInstabR, to obtain the relative importance of the factors that affect species distribution based on the concept of stability.

## Results

### Environmental layer

A global environmental layer was generated with each of the variables, using information from WorldClim (Fig 2) and a local layer with data from weather stations and Marksim®, to enhance the distribution model (Fig 3). The local layer contains variables BIO1, BIO8, BIO12, BIO13, BIO14, VI, Slope, Altitude, TH24 (with 5x5 cells and in current conditions) and PP. Presence maps for *Diatraea* spp. were generated at global, national and local scales (Fig 4).

**Table 1. WorldClim bioclimatic variables.**

| Code | Variable |
| --- | --- |
| BIO1 | Annual Mean Temperature |
| BIO8 | Mean Temperature of Wettest Quarter |
| BIO12 | Annual Precipitation |
| BIO13 | Precipitation of Wettest Month |
| BIO14 | Precipitation of Driest Month |
| BIO19 | Precipitation of Coldest Quarter. |
| VI | Normalized Vegetation Index |
| Slope | Topographic slope (˚) |
| Altitude | Altitude (masl) |
| TH24 | Topographical heterogeneity, 24 surrounding cells with A 5' x 5 |
| PP | Primary Production |

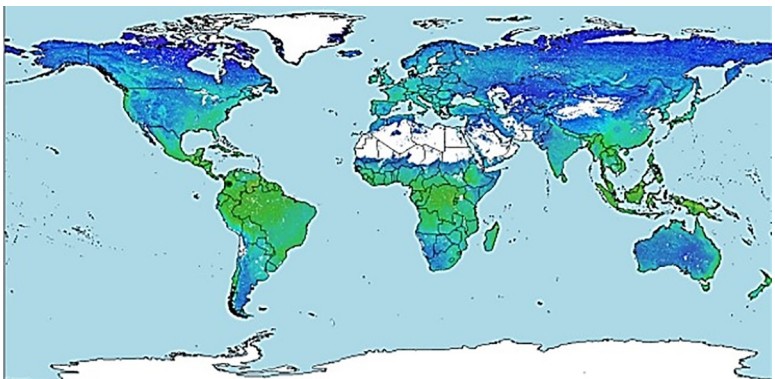

**Fig 2. Global-scale environmental layer using WorldClim.** Colors indicate the mixture of BIO1, BIO8, BIO12, BIO13, BIO14, VI, Slope, Altitude, TH24, PP for each zone in a range of 5'x5' cell resolution. Green indicates areas of potential presence of *Diatraea* spp. and blue indicates areas whose environmental conditions do not guarantee insect´s life cycle.

## Variable contribution

The effect of environmental variables on *Diatraea* spp. distribution shows direct incidence of BIO12, VI, BIO19, PP, Slope and BIO1. Although temperature is considered a main factor affecting insect physiology, precipitation and its association with cloudy conditions can alter navigation in the nocturnal adult stage, thus becoming a determining factor in movement and distribution in a given range. Food availability throughout the year and the type of topography are also definite limiting factors. In this analysis altitude did not show up as a limiting factor for pest distribution (Fig 5).

## Optimum niche

The analysis of optimal niche estimates indicated that *Diatraea* spp. may express their maximum adaptive potential under temperatures between 20°C and 23°C, accumulated annual

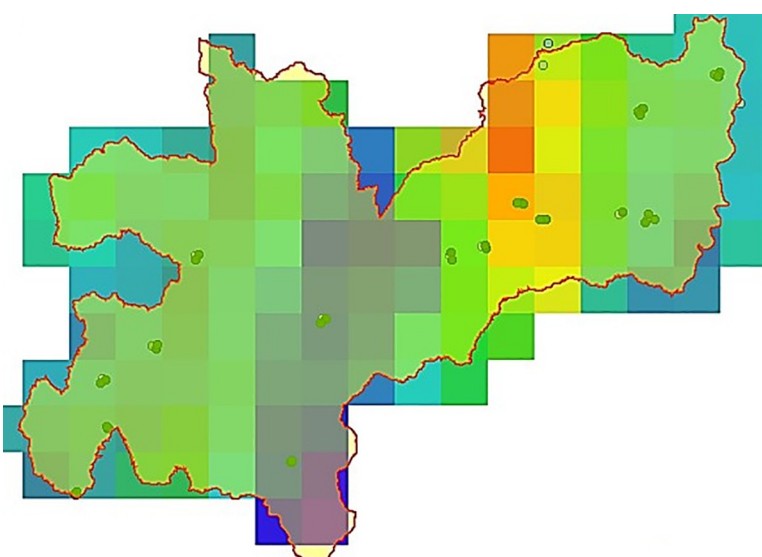

**Fig 3. Environmental layer for Caldas department, Colombia.** Range of 5'x5' cell resolution. Dots indicate records of *Diatraea* spp. Areas in green, red or yellow have a high probability of having presence of *Diatraea* spp. Other colors indicate areas where the insect could be present, but with smaller populations.

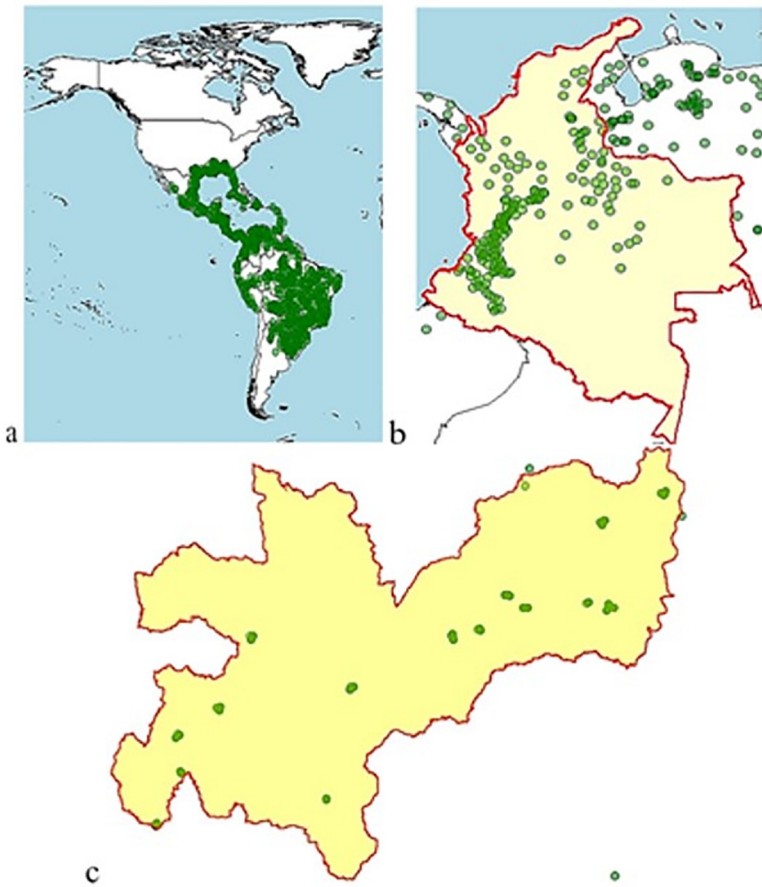

**Fig 4. Presence points of *Diatraea* spp. at different scales.** a) Western Hemisphere, b) Colombia and c) Department of Caldas. Dots indicate records of presence.

rainfall between 1200 and 1500 mm, dry months with precipitations below 50 mm, slopes below 0.05 degrees, crop heterogeneity with an index of 0.2 and primary production values of 1.0. These conditions are similar to those present throughout the year in the CRV and the

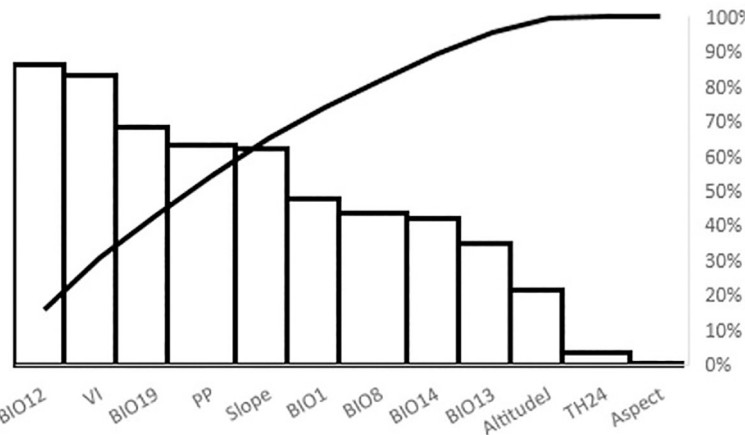

**Fig 5. Percentage contribution of environmental variables to *Diatraea* spp. distribution.**

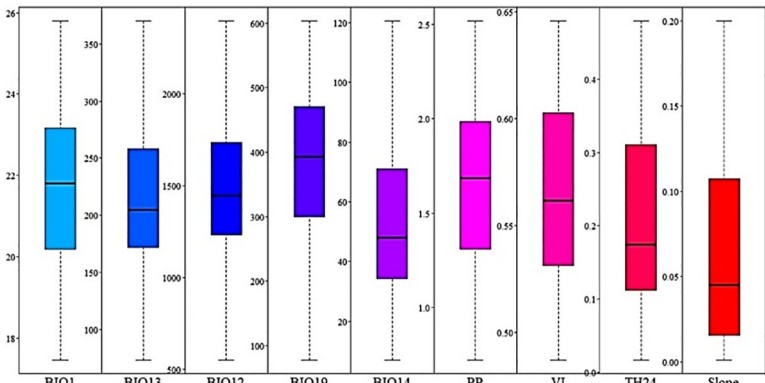

**Fig 6. Estimate of an optimum niche for the establishment of *Diatraea* spp.** In the model, quartiles indicate areas with environmental conditions favorable for development of *Diatraea* spp., with those under 50% representing optimum conditions.

areas sampled in the department of Caldas. It is noteworthy that the ranges obtained here may have a greater or lesser amplitude for each environmental variable, facilitating the mobilization of pest populations towards areas different from those originally occupied (Fig 6).

### *Diatraea* spp. richness

The most notable differences among sites, determining flora and fauna distribution, were related to soil conditions, topography, altitude, ambient temperature and rainfall. The matrix analyzed here identified factors and their relative contribution to species richness. The negative effect of anthropogenic factors such as primary production (PP) and vegetation index (VI)—even without taking into account effects associated with pest management that were not considered here—reduced species richness in several potential areas (Fig 7). Polar Coordinates (Eq 2) suggest high probability of *Diatraea* spp. dispersion associated to precipitation and slope as the determinant environmental factors in the movement of adults to colonize new areas (Fig 7).

$$x_{n+1} = x_n + \sin\,(angle\ x\ distance)\quad y_{n+1} = y_n + \cos\,(angle\ x\ distance) \tag{2}$$

### Distribution maps

Indications of a possible reduction in the number of niches and therefore in the population of *Diatraea* spp. are given by the BCC-CSM1-1 climate change model, projected for a period of 50 years, under the premise of an increase in $CO_2$ and greenhouse gas concentrations, at the same time that Earth temperatures increase, impacting life cycles and dispersion due to meteorological alterations and changes in the ecosystems (Fig 8). The impact is greater when effects are projected over 70 years, and where $CO_2$ and greenhouse gas concentrations are projected to increase global temperature by as much as 1˚C. The estimate is that under these conditions, *Diatraea* spp. populations would be reduced by 50% in their current and potential niches (Fig 9).

In Colombia, climate change is expected to affect the dispersion of *Diatraea* spp. populations, forcing populations to concentrate in western Colombia and the Caribbean, and changing pest distribution (currently predominant in southern, central and western parts of the country). Niche decrease may be greater than 50%, when the climate change model is projected over a 50-year period (Fig 10). It is important to note that even though the *Diatraea* pest complex is a well-recognized sugarcane pest, it is also found in corn, rice, sorghum and some forage grasses [17]. In addition, models of maximum entropy and climate change show

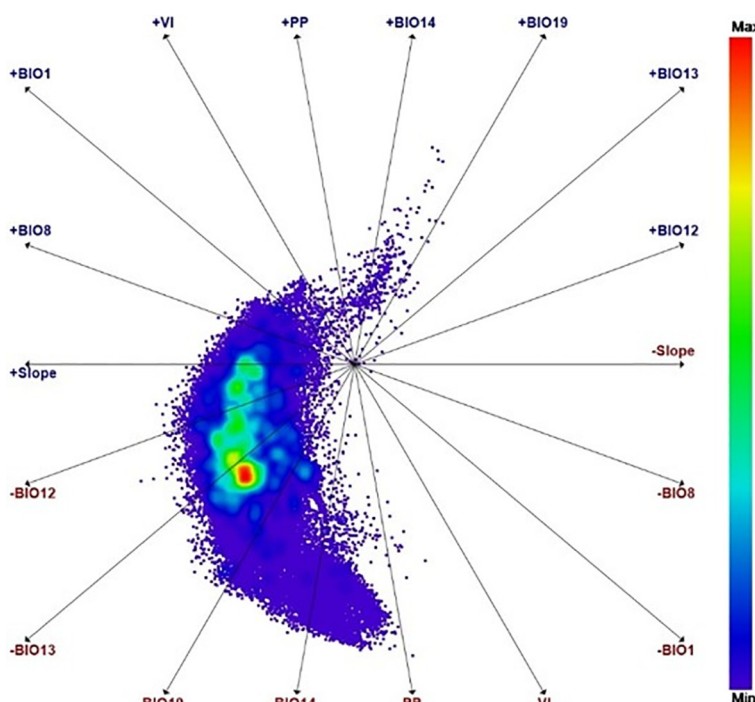

**Fig 7. Polar coordinates to evaluate the importance of environmental variables in the distribution of *Diatraea* spp.** BIO1, BIO8, BIO10, BIO12, BIO12, VI, PP, BIO13 and TH24.

reductions of more than 70% in pest populations as a result of changes in water regimes and winds, radically changing climate dynamics at different scales (macro, meso and micro) (Fig 11). The three departments analyzed (Caldas, Risaralda and Valle del Cauca) in western Colombia exhibit a notorious reduction of niches and consequently of populations, but sustain current presence of the insect's populations in some areas. The highest concentration and production of sugarcane in the country is located in the CRV. Therefore, it is very likely that this region will sustain resources (i.e., plants to feed upon) for *Diatraea busckella* populations to prosper during and beyond the next 50 years (Fig 12).

## Discussion

Climate data is widely used to model potential geographic species distribution [29, 30] as it allows the use of a greater number of historical records on the target species and does not present inconsistencies between records and shapes used to model. In this way, temporal shapes—such as vegetation, whose correspondence with target species would require updated information on the temporal availability of that shape—can be substituted.

The broadening of Geographic Information Systems and the growth of applied statistical techniques have allowed the expansion of tools for spatial analysis and species distribution studies [31–36]. Species distribution models are not only useful to evaluate possible effects of environmental factors on natural ecosystems, but also on agricultural ones, as well as to highlight possible population changes on different species commonly associated to these areas, anticipating pest outbreaks and/or new colonizers.

Initial models described the relationship between climatic phenomena, insect distribution and behavior through linear relationships [37–44], assuming the area under the curve as a good representation of the relation between developmental rate and temperature, as in the

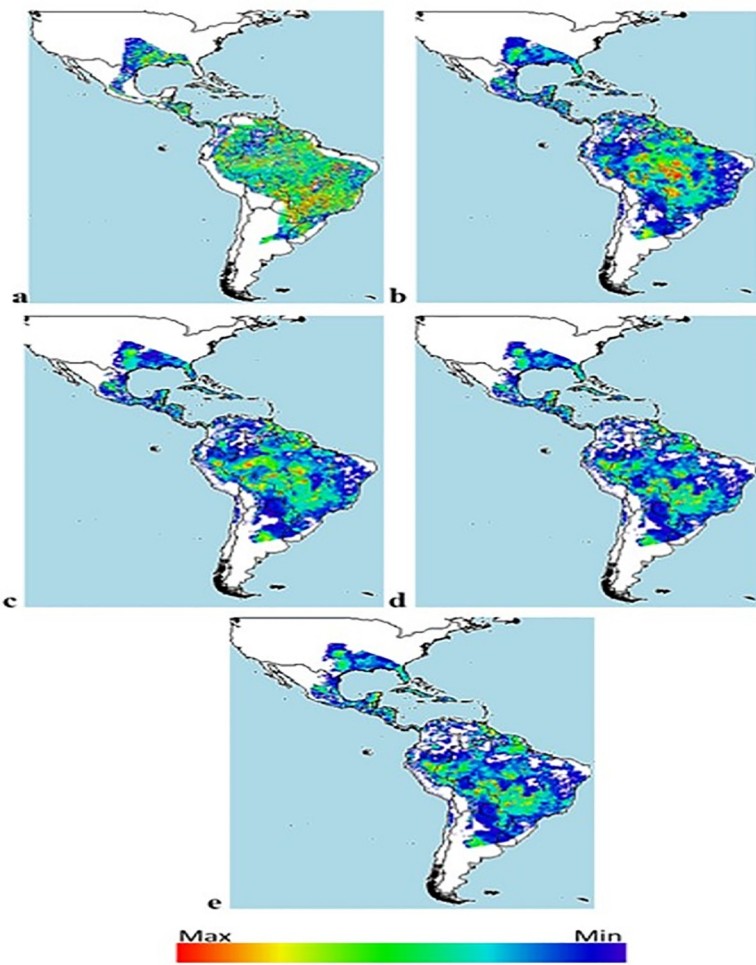

**Fig 8. Distribution of *Diatraea* spp. in the Western Hemisphere under different climate change (BCC-CSM1-1) and carbon concentration projections over 50 years.** a) current condition, b) rcp26, c) rcp45, d) rcp60, e) rcp85. Color indicates maximum or minimum presence in a likely niche and under favorable conditions.

case of asymmetric, exponential and logistic models [45–47]. In studies on *Diatraea saccharalis* F. spatial distribution, in order to propose changes in scouting [48], data was adjusted by Poisson distribution, assuming that all individuals have the same probability of occupying a place in any given space [12], but also using negative binomial distribution, in which occurrence of individuals limits the presence of others [49]. In our case linear and maximum entropy models were used [50–56] to improve estimates on minimum developmental threshold.

Descriptions of current exotic flora and fauna of Great Britain reflect a small subset of resident species in climatologically comparable regions for which important links were established [57, 58]. Therefore, the way in which climate change can influence future pathways for possible exotic species needs to be understood, considering that natural dynamics can also be affected by global warming due to alterations in the accumulation of day-grades for each physiological process.

Given its tropical location, Colombia is affected by patterns of either bimodal (Andean region) or unimodal rainfall patterns (eastern Colombia), when El Niño or La Niña conditions are not present [59]. These patterns determine insect distribution and environmentally-

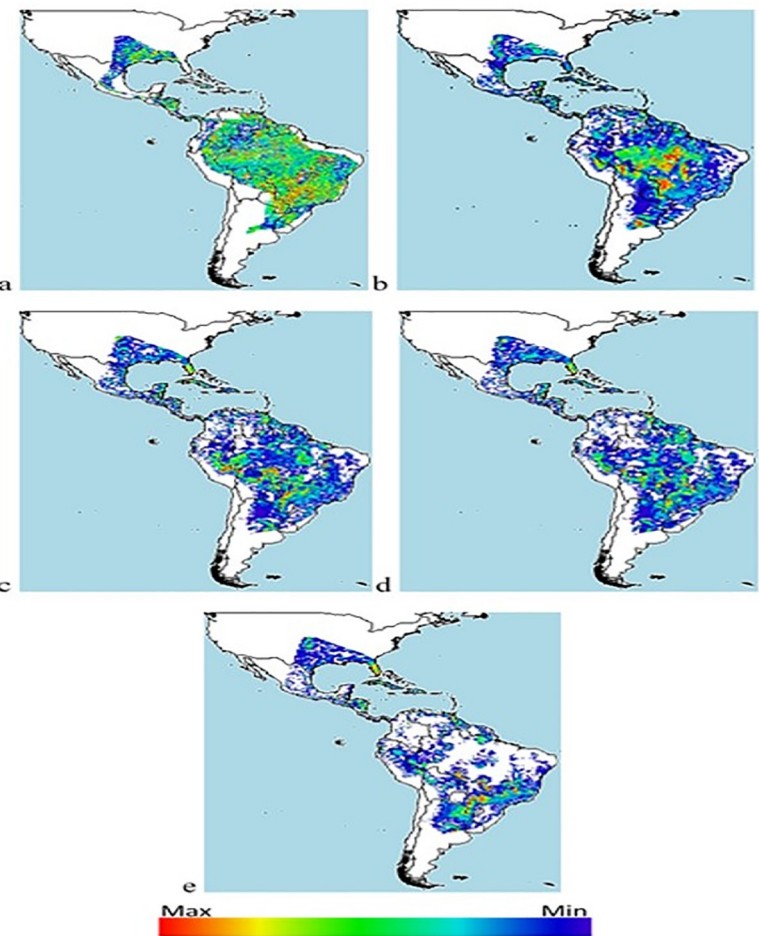

**Fig 9. Distribution of *Diatraea* spp. in the Western Hemisphere under different climate change (BCC-CSM1-1) and carbon concentration projections over 70 years.** a) current condition, b) rcp26, c) rcp45, d) rcp60, e) rcp85. Color indicates maximum or minimum presence in a likely niche and under favorable conditions.

optimal zones for population development. Invertebrates are particularly sensitive to climatic conditions, and parameters such as temperature, precipitation, relative humidity and soil moisture are useful in predicting important events in the growth of pest populations [13, 60].

Our analysis suggests that *Diatraea* species can colonize new areas, specifically during the drier months, when precipitation is not a limiting factor for adult distribution, and when threshold temperatures (ranging from 18 to 26˚C) guarantee full development and determine successful establishment as food availability is assured (i.e., sugarcane grown all year round). In this regard and associated with climate change, changes have been determined in pest profiles in South Australia [61], changes in the distribution of aphid flights have been noted across Europe [62], and a destabilization of the tea moth *Adoxophyes honmai* Yasuda (Lepidoptera: Tortricidae) outbreak cycles in Japan has been highlighted [63]; demonstrating that due to their rapid development arthropods are essential to underscore climate changes environmental impacts at a global scale.

The main expectation concerning recent and projected global climate variability scenarios will influence the likelihood of species improving the use of available thermal energy for growth and reproduction. Our analysis of *Diatraea* species distribution shows a possible

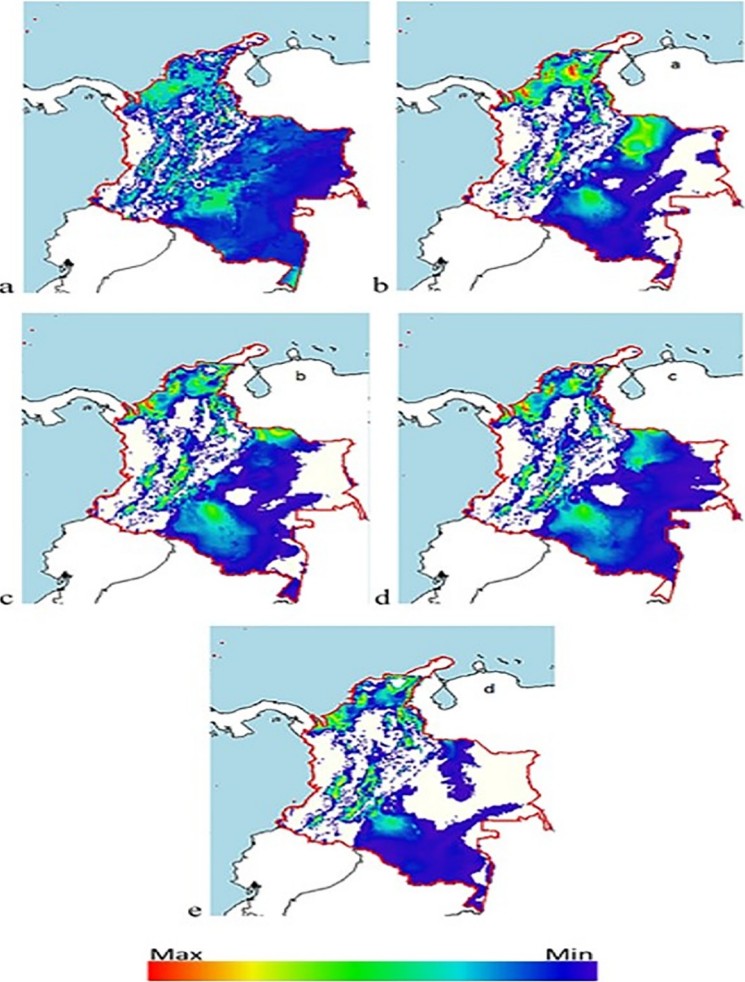

**Fig 10. Distribution of *Diatraea* spp. in Colombia under different climate change (BCC-CSM1-1) and carbon concentration projections over 50 years.** a) current condition, b) rcp26, c) rcp45, d) rcp60, e) rcp85. Color indicates maximum or minimum presence in a likely niche and under favorable conditions.

reduction in the number of niches and therefore in the populations of *Diatraea* spp. at a global, national and regional scales, with greater impacts when effects are projected over 70 years. In agreement to what we found in the *Diatraea* case, the distribution of the three *Penthaleus* species (Acari: Penthaleidae) in Australia is correlated with different climate variables and suitable climatic conditions are likely to decrease in the future [64]; but in general positive associations between climate change and pests increases have been reported; such as the expectation of generalized increases in fungal, plant and arthropod communities in Great Britain [65]; the expansion of the suitable habitat area for the Colorado potato beetle *Leptinotarsa decemlineata* (Coleoptera: Chrysomelidae) and the European corn borer *Ostrinia nubilalis* (Lepidoptera: Crambidae) in central Europe [66]; risks of additional generations of the apple moth *Cydia pomonella* (Lepidoptera: Tortricidae) in Switzerland [67]; and invasive populations of the African fig fly *Zaprionus indianus* (Diptera: Drosophilidae) colonizing India, after latitudinal clones of rapid adaptation [68]. In general, changes in phenology and voltinism will require a change in plant protection strategies but, as illustrated in our work, this will depend on the

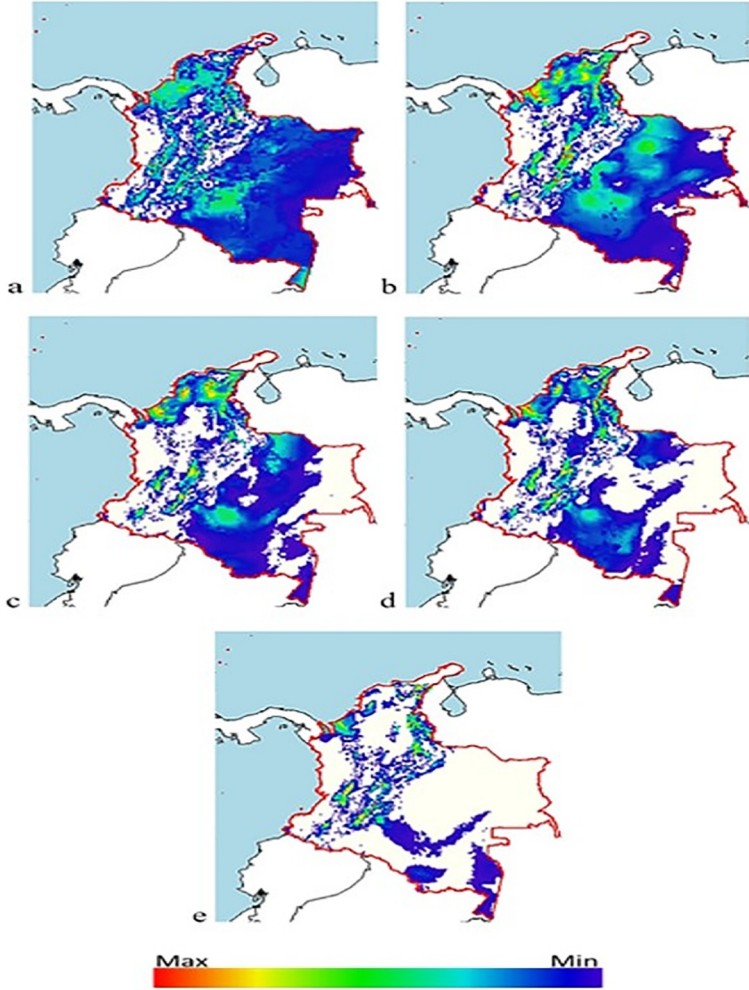

**Fig 11. Distribution of *Diatraea* spp. in Colombia under different climate change (BCC-CSM1-1) and carbon concentration projections over 70 years.** a) current condition, b) rcp26, c) rcp45, d) rcp60, e) rcp85. The color chart indicates maximum or minimum presence in a likely niche under favorable conditions.

system under study and the conditions, which may benefit some species but be deleterious for others.

## Concluding remarks

Species distribution models, based on statistical and cartographic procedures plus real presence data, have become a tool for analysis of organisms spatial patterns, making it possible to infer potentially suitable areas based on favorable environmental conditions. In this sense, data from natural history collections acquires a new use. Information about species distribution is increasingly important for a wide range of aspects of decision support and management tactics, such as biodiversity studies, species protection, species reintroduction, exotic species, prediction of potential effects of ecosystem loss or global climate change, etc. Such information is crucial in constructing species distribution models, as deficiencies in data quality lead to uncertainty in such models.

*Diatraea* spp. is strongly influenced by the effects of climate change, not only at a global and regional scale, but also at the local scale. Under the environmental models evaluated for

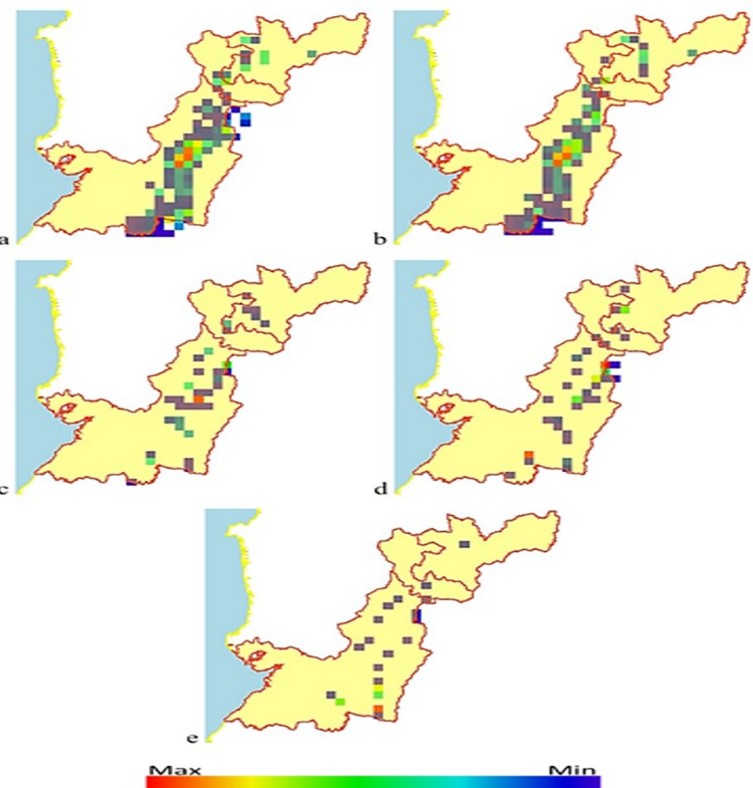

**Fig 12. Distribution of *Diatraea busckella* under different climate change projections (BCC-CSM1-1) over 50 years in the departments of Caldas (top), Risaralda (middle) and Valle del Cauca (bottom) in southwestern Colombia.** a) current condition, b) rcp26, c) rcp45, d) rcp60, e) rcp85. Color indicates maximum or minimum presence in a likely niche and under favorable conditions.

the next 50 to 70 years, climate change will reduce the species' population niches as well as the number of individuals, with reductions that may exceed 70% in comparison to current records. An optimal niche for the *Diatraea* species is estimated to require temperatures between 20˚C and 23˚C, accumulated annual rainfall between 1200 and 1500 mm, dry months with precipitations below 50 mm, slopes below 0.05 degrees, crop heterogeneity with an index of 0.2 and primary production values of 1.0. These conditions are similar to those present throughout the year in the CRV and the municipalities sampled in the Caldas department.

The present research allows us to give a future approximation on the incidence of a pest complex that is a mayor limiting factor for sugarcane production, but that also uses alternate hosts such as rice, corn and other gramineous plants. The species may be present during global warming scenarios, with reduced populations, but remaining in currently inhabited spaces and colonizing new places, where conditions are optimal for development. The development of these models is intended to feed support systems for decision-making and reduce the levels of uncertainty when looking for new management and pest control practices and for establishing new sugarcane projects in the country.

## Acknowledgments

The authors would like to express their appreciation to the Colombian Sugarcane Research Center (CENICAÑA) for its valuable contribution and collaboration in the supply, development, and analysis of information; to the BEKDAU Center for Research, Innovation and Technology of the sugarcane sector in the Caldas Department for contributing significantly to the development and execution of the project; and to Washington State University (WSU) for its guidance in the development and analysis of each of the parameters evaluated. We also thank Alexandra Walter for valuable editorial assistance.

## Author Contributions

**Conceptualization:** Alberto Soto Giraldo, Luis Fernando Vallejo Espinosa.

**Data curation:** Gabriel Jaime Castaño Villa.

**Formal analysis:** Gabriel Jaime Castaño Villa, Melba Ruth Salazar Gutierrez.

**Investigation:** Julián Andrés Valencia Arbeláez.

**Methodology:** Julián Andrés Valencia Arbeláez, Luis Fernando Vallejo Espinosa.

**Project administration:** Alberto Soto Giraldo.

**Resources:** Alberto Soto Giraldo.

**Software:** Julián Andrés Valencia Arbeláez.

**Supervision:** Alberto Soto Giraldo, Gabriel Jaime Castaño Villa, Luis Fernando Vallejo Espinosa.

**Validation:** Gabriel Jaime Castaño Villa, Melba Ruth Salazar Gutierrez, Germán Vargas.

**Writing – original draft:** Julián Andrés Valencia Arbeláez.

**Writing – review & editing:** Alberto Soto Giraldo, Gabriel Jaime Castaño Villa, Melba Ruth Salazar Gutierrez, Germán Vargas.

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
