## [Decision Letter · Decision Letter 0]

13 Aug 2020

PONE-D-20-18805

POPULATION DYNAMICS OF Diatraea spp. UNDER DIFFERENT CLIMATE OFFER CONDITIONS IN COLOMBIA.

PLOS ONE

Dear Dr. Valencia Arbeláez,

Thank you for submitting your manuscript to PLOS ONE. After careful consideration, we feel that it has merit but does not fully meet PLOS ONE’s publication criteria as it currently stands. Therefore, we invite you to submit a revised version of the manuscript that addresses the points raised during the review process.

We look forward to receiving your revised manuscript.

Kind regards,

Shamsuddin Shahid

Academic Editor

PLOS ONE

Additional Editor Comments:

Both the reviewers think the paper is interesting. But both of them asked for more description of methods and justification of the data and methods used in the study. I also believe that significant improvement in description of methods is required.

Journal Requirements:

3. During your revisions, please note that a simple title correction is required: to follow correct English language usage, the title should read"Population dynamics of Diatraea spp. under different climate scenarios in Colombia". Please ensure this is updated in the manuscript file and the online submission information.

"Unfunded studies"

6. We note that Figures in your submission contain map images which may be copyrighted. All PLOS content is published under the Creative Commons Attribution License (CC BY 4.0), which means that the manuscript, images, and Supporting Information files will be freely available online, and any third party is permitted to access, download, copy, distribute, and use these materials in any way, even commercially, with proper attribution. For these reasons, we cannot publish previously copyrighted maps or satellite images created using proprietary data, such as Google software (Google Maps, Street View, and Earth). For more information, see our copyright guidelines: http://journals.plos.org/plosone/s/licenses-and-copyright.

6.1.    You may seek permission from the original copyright holder of Figures to publish the content specifically under the CC BY 4.0 license. 

6.2.    If you are unable to obtain permission from the original copyright holder to publish these figures under the CC BY 4.0 license or if the copyright holder’s requirements are incompatible with the CC BY 4.0 license, please either i) remove the figure or ii) supply a replacement figure that complies with the CC BY 4.0 license. Please check copyright information on all replacement figures and update the figure caption with source information. If applicable, please specify in the figure caption text when a figure is similar but not identical to the original image and is therefore for illustrative purposes only.

7. Please upload copies of Figures 7-12, to which you refer in your text. If the figures are no longer to be included as part of the submission please remove all reference to them within the text.

Reviewers' comments:

Reviewer's Responses to Questions

**Comments to the Author**

1. Is the manuscript technically sound, and do the data support the conclusions?

Reviewer #1: Yes

Reviewer #2: Yes

2. Has the statistical analysis been performed appropriately and rigorously? 

Reviewer #1: Yes

Reviewer #2: Yes

3. Have the authors made all data underlying the findings in their manuscript fully available?

Reviewer #1: No

Reviewer #2: Yes

4. Is the manuscript presented in an intelligible fashion and written in standard English?

Reviewer #1: No

Reviewer #2: Yes

5. Review Comments to the Author

Reviewer #1: Comments for Authors

The manuscript “population dynamics of Diatraea spp. under different climate offer

conditions in Colombia” present an interesting study on how climate change is expected to affect the distribution of sugar cane stem borers, Diatraea spp. The manuscript is well structured and nicely presented. However, I have some observations and comments for the authors as follows.

Abstract

Authors should mention for which scenarios the study was conducted for.

Authors should include more key findings from the study in the abstract.

Introduction

1. The Earth's temperature increased between 0. 7°C from 1900, 1. 3°C from 1950 and 1. 8°C

over the past 35 years. This sentence should be rephrased for better understanding for readers.

2. Many of the sentences in this section are too long; authors should make them shorter or split them.

Materials and Methods

1. Change ‘The climate change model’ to ‘The global climate model (GCM)’

2. Why was this BCC_CSM1 chosen?

3. Change (CPR) to (RCP)

4. Change ‘depending on the number’ to ‘depending on the amount’

5. Remove ‘it’ in the sentence: MODESTR®, like MAXENT®, (Elith et al., 2011, Phillips and Dudik, 2008) it

6. Authors should split the paragraph that started with: To generate the distribution maps and to see the effect for each variable in the distribution of Diatraea spp., MODESTR® (http://www.ipezes/ModestR)..........

Results

1. Why are legends not given in the Figures in the results?

2. Authors should label the axis of Figure 5.

3. Label axis of Figure 6.

4. The authors mentioned anthropogenic factors affecting the Diatraea spp. Under the section: Factors affecting Diatraea spp. Richness. Are there other anthropogenic factors aside from the primary production (PP) and vegetation index e.g. does factors such as pest control affect them?

5. Under section distribution maps: Change this word ‘affection’ to ‘impaction’

6. From section 3.4 authors should check the figure numberings and correct them.

Other comments

Authors should include lines numbering in manuscript the next times

There are grammatical and typographical errors in the manuscript. Authors should check and correct for these.

Reviewer #2: POPULATION DYNAMICS OF Diatraea spp. UNDER DIFFERENT CLIMATE OFFER CONDITIONS IN COLOMBIA.

The authors applied for the first time a species distribution modelling approach to study Diatraea species. They estimated potential distributions and niche overlap using modelling software. Results suggests

that these species could be affected by climate change and that they have distinct ecological niches. They propose SDMs as a cost-effective tool to explore ecological and biological limits of non-model species and inform management decisions on environmental issues like climate change.

The study is overall interesting and extends our understanding of ecology and biology of insect populations. The proposed approach could be useful for other studies focusing on the impact of climate change on species distribution and their conservation/management. However, some conceptual and methodological details are not fully clear, and several

points need to be elucidated. Please find additional details in the specific comments below.

Abstract: Describe more generally your results and conclusion.

figures could benefit from a shorter title.

6. PLOS authors have the option to publish the peer review history of their article (what does this mean?). If published, this will include your full peer review and any attached files.

Reviewer #1: No

Reviewer #2: **Yes: **Saeed Mohammadi

---

## [Author Response · Author response to Decision Letter 0]

25 Nov 2020

PlosOne

Response to Reviewers

Journal Requirements

PLOS ONES´s style requirements have been included

 We suggest you thoroughly copyedit your manuscript for language usage, spelling, and grammar. If you do not know anyone who can help you do this, you may wish to consider employing a professional scientific editing service. 

We hired Alexandra Walter, a professional Spanish-English translator with 30 years of experience with scientific publications at the International Center for Tropical Agriculture in Colombia - CIAT. Alexandra retired from CIAT in 2008 and works as a freelance editor.

 We note that Figures in your submission contain map images which may be copyrighted. All PLOS content is published under the Creative Commons Attribution License (CC BY 4.0), which means that the manuscript, images, and Supporting Information files will be freely available online, and any third party is permitted to access, download, copy, distribute, and use these materials in any way, even commercially, with proper attribution. For these reasons, we cannot publish previously copyrighted maps or satellite images created using proprietary data, such as Google software (Google Maps, Street View, and Earth). For more information, see our copyright guidelines: http://journals.plos.org/plosone/s/licenses-and-copyright.

We used ModestR to design the maps, so they are original and not previously copyrighted. Fig 1 is a free available figure from the Cenicaña´s webpage whose source is accordingly cited.

 Please provide a direct link to the source of the replacement map in Figure 1.

The distribution map of the weather stations was generated by QGIS 3.6.1, according to coordinates of the different stations. As the map was made using free software, there is no direct link to share.

 We note that your Financial Disclosure statement is currently as follows: "Unfunded studies".

The Project was funded through resources from by the Colombian Institute for the Development of Science and Technology (COLCIENCIAS) by grant program number 647, in addition to support through the use of infrastructure and laboratories of the Colombian Sugarcane Research Center (Centro de Investigación de la Caña de Azúcar de Colombia, Cenicaña)

Comments for Authors

The manuscript “population dynamics of Diatraea spp. under different climate offer

conditions in Colombia” present an interesting study on how climate change is expected to affect the distribution of sugar cane stem borers, Diatraea spp. The manuscript is well structured and nicely presented. However, I have some observations and comments for the authors as follows.

Abstract

Authors should mention for which scenarios the study was conducted for. 

Done

Authors should include more key findings from the study in the abstract. 

More key findings have been included in the abstract

Introduction

 The Earth's temperature increased between 0. 7°C from 1900, 1. 3°C from 1950 and 1. 8°C

over the past 35 years. This sentence should be rephrased for better understanding for readers. 

Sentence has been rephrased for better understanding

 Many of the sentences in this section are too long; authors should make them shorter or split them. 

Sentences have been edited and organized for better understanding

Materials and Methods

 Change ‘The climate change model’ to ‘The global climate model (GCM)’ 

Done

 Why was this BCC_CSM1 chosen?

Lines 138 – 141: “… according to the Colombian Institute of Hydrology, Meteorology and Environmental Studies (Instituto de Hidrología, Meteorología y Estudios Ambientales, IDEAM) this model fits well Colombian meteorological conditions.”

 Change (CPR) to (RCP) 

Done

 Change ‘depending on the number’ to ‘depending on the amount’

Done

 Remove ‘it’ in the sentence: MODESTR®, like MAXENT®, (Elith et al., 2011, Phillips and Dudik, 2008) it

Done

 Authors should split the paragraph that started with: To generate the distribution maps and to see the effect for each variable in the distribution of Diatraea spp., MODESTR® (http://www.ipezes/ModestR)..........

Done

Results 

 Why are legends not given in the Figures in the results? 

Legends have been added to figs

 Authors should label the axis of Figure 5. 

Done

 Label axis of Figure 6.

Done

 The authors mentioned anthropogenic factors affecting the Diatraea spp. Under the section: Factors affecting Diatraea spp. Richness. Are there other anthropogenic factors aside from the primary production (PP) and vegetation index e.g. does factors such as pest control affect them?

Pest control tactics have the potential to affect directly insect populations. However, no detailed information is available on control programs in the areas under study, being the case that in some of those no control is executed, so we did no considered this factor and rather focus on making a first distribution approach in relation to environmental, topographical and agricultural factors.

 Under section distribution maps: Change this word ‘affection’ to ‘impaction’

Done

 From section 3.4 authors should check the figure numberings and correct them. 

Done

Other comments

Authors should include lines numbering in manuscript the next times

Done

There are grammatical and typographical errors in the manuscript. Authors should check and correct for these. 

We hired a professional scientific editing service

PDF Comments

Abstract

 Does not need to say this here in the abstract

Done

Introduction

 Is true it effects on just these areas??

Lines 90-93 “To understand how climate change can affect sugarcane borers distribution, predictive inter- and intra-annual scale models were applied at the global, national and local scales, aiming to develop a framework to guide pest management decisions, thus reducing levels of uncertainty in the face of different climate scenarios. “

Materials and methods

 you need to sort a new section regarding the study genus and introduce it including the number of species...

Lines 98 – 106 “whereas in the departments of Risaralda and Valle del Cauca, samples were collected from 2010 to 2017 from different municipalities in what it is known as the Cauca River Valley (CRV), a region that extends from northern Cauca to southern Risaralda (Fig 1). Larvae were collected in each sugarcane plot by a uniform 2 man-hour sampling effort, following the linear distribution of crop rows. Larvae were then taken to the laboratory and identified following taxonomic keys [17]. In addition, this information was used in combination with the CABI - Invasive Species Compendium (www.cabi.org) – to increase data on distribution of Diatraea spp. and increasing modeling accuracy”.

 you need to explain how measure these and their source?

Lines 116-131 “/). At the same time, open-access weather portals were consulted such as WorldClim, from which 19 bioclimatic variables were collected on monthly rainfall and temperature [18]. The variance inflation factor (VIF) was calculated to analyze correlations between variables [19,20], providing an index that measures the extent to which the variance (the square of the estimated standard deviation) increases due to the collinearity, which allows for discarding climatic variables that could generate noise through redundancy.

Following the VIF analysis, only 6 of the 19 variables provided by WorldClim were used, plus slope (topographic slope in degrees), slope-aspect (which is defined as the compass direction to which a slope faces measured in degrees), altitude (meters), vegetation index (VI), terrestrial primary production (TPP, gCm2 d1), and TH24 (topographic heterogeneity calculated for the 24 surrounding cells) [21,22], and used along with those supplied within the ModestR® software

 change the title of figure 3.

Done

 seems you explained before these abbreviations in the table 1. Do not need to repeat again.

Done

 please say more details in Methods for this approach, polar coordinates

To find that, we use the following ecuation 

x_(n+1)=x_n+sin⁡(angle x distance) y_(n+1)=y_n+cos⁡(angle x distance)

 where? which country?

Colombia

Conclusions 

 does not make sense when you did not compare different softwares! you applied just this, so delete this part.

Done

---

## [Decision Letter · Decision Letter 1]

15 Dec 2020

Population dynamics of sugarcane borers, Diatraea spp., under different climatic scenarios in Colombia

PONE-D-20-18805R1

Dear Dr. Valencia Arbeláez,

We’re pleased to inform you that your manuscript has been judged scientifically suitable for publication and will be formally accepted for publication once it meets all outstanding technical requirements.

Kind regards,

Shamsuddin Shahid

Academic Editor

PLOS ONE

Additional Editor Comments (optional):

Reviewers' comments:

Reviewer's Responses to Questions

**Comments to the Author**

1. If the authors have adequately addressed your comments raised in a previous round of review and you feel that this manuscript is now acceptable for publication, you may indicate that here to bypass the “Comments to the Author” section, enter your conflict of interest statement in the “Confidential to Editor” section, and submit your "Accept" recommendation.

Reviewer #1: All comments have been addressed

Reviewer #2: All comments have been addressed

2. Is the manuscript technically sound, and do the data support the conclusions?

Reviewer #1: Yes

Reviewer #2: Yes

3. Has the statistical analysis been performed appropriately and rigorously? 

Reviewer #1: Yes

Reviewer #2: Yes

4. Have the authors made all data underlying the findings in their manuscript fully available?

Reviewer #1: Yes

Reviewer #2: Yes

5. Is the manuscript presented in an intelligible fashion and written in standard English?

Reviewer #1: Yes

Reviewer #2: Yes

6. Review Comments to the Author

Reviewer #1: Authors have appropraitely addressed all comments. Manuscript is technically sound and presents an interesting research on the study topic.

Reviewer #2: You did not mention anything about the biodiversity values of studied species? It would be very good to add more details.

7. PLOS authors have the option to publish the peer review history of their article (what does this mean?). If published, this will include your full peer review and any attached files.

Reviewer #1: No

Reviewer #2: **Yes: **Saeed Mohammadi- Department of Environmental Sciences, University of Zabol, Zabol, Iran

---

## [Editor Report · Acceptance letter]

2 Jan 2021

PONE-D-20-18805R1 

Population dynamics of sugarcane borers, *Diatraea* spp., under different climatic scenarios in Colombia 

Dear Dr. Valencia Arbeláez:

I'm pleased to inform you that your manuscript has been deemed suitable for publication in PLOS ONE. Congratulations! Your manuscript is now with our production department. 

Kind regards, 

on behalf of

Dr. Shamsuddin Shahid 

Academic Editor

PLOS ONE